# Structures of SALSA/DMBT1 SRCR domains reveal the conserved ligand-binding mechanism of the ancient SRCR fold

Martin P Reichhardt[1], Vuokko Loimaranta[2], Susan M Lea[1,3], Steven Johnson[1]

The scavenger receptor cysteine-rich (SRCR) family of proteins comprises more than 20 membrane-associated and secreted molecules. Characterised by the presence of one or more copies of the ~110 amino-acid SRCR domain, this class of proteins have widespread functions as antimicrobial molecules, scavenger receptors, and signalling receptors. Despite the high level of structural conservation of SRCR domains, no unifying mechanism for ligand interaction has been described. The SRCR protein SALSA, also known as DMBT1/gp340, is a key player in mucosal immunology. Based on detailed structural data of SALSA SRCR domains 1 and 8, we here reveal a novel universal ligand-binding mechanism for SALSA ligands. The binding interface incorporates a dual cation-binding site, which is highly conserved across the SRCR superfamily. Along with the well-described cation dependency on most SRCR domain–ligand interactions, our data suggest that the binding mechanism described for the SALSA SRCR domains is applicable to all SRCR domains. We thus propose to have identified in SALSA a conserved functional mechanism for the SRCR class of proteins.

## Introduction

The salivary scavenger and agglutinin (SALSA), also known as gp340, "deleted in malignant brain tumors 1" (DMBT1) and salivary agglutinin (SAG), is a multifunctional molecule found in high abundance on human mucosal surfaces (1, 2, 3, 4). SALSA has widespread functions in innate immunity, inflammation, epithelial homeostasis, and tumour suppression (5, 6, 7). SALSA binds and agglutinates a broad spectrum of pathogens including, but not limited to, human immunodeficiency virus type 1, *Helicobacter pylori*, *Salmonella enterica* serovar Typhimurium, and many types of streptococci (8, 9, 10, 11). In addition to its microbial scavenging function, SALSA has been suggested to interact with a wide array of endogenous immune defence molecules. These include secretory IgA, surfactant proteins A (SP-A) and D (SP-D), lactoferrin, mucin-5B, and components of the complement system (1, 2, 12, 13, 14, 15, 16, 17, 18). SALSA thus engages innate immune defence molecules and has been suggested to cooperatively mediate microbial clearance and maintenance of the integrity of the mucosal barrier.

The 300- to 400-kD SALSA glycoprotein is encoded by the *DMBT1* gene. The canonical form of the gene encodes 13 highly conserved scavenger receptor cysteine-rich (SRCR) domains, followed by two C1r/C1s, urchin embryonic growth factor and bone morphogenetic protein-1 (CUB) domains that surround a 14th SRCR domain, and finally a zona pellucida domain at the C terminus (19, 20). The first 13 SRCRs are 109 aa domains found as "pearls on a string" separated by SRCR-interspersed domains (SIDs) (Fig 1A) (1, 21). The SIDs are 20- to 23-aa-long stretches of predicted disorder containing a number of glycosylation sites, which have been proposed to force them into an extended conformation of roughly 7 nm (7). In addition to this main form, alternative splicing and copy number variation mechanisms lead to expression of variants of SALSA containing variable numbers of SRCR domains in the N-terminal region.

The SRCR protein superfamily include a range of secreted and membrane-associated molecules, all containing one or more SRCR domains. For a number of these molecules, the SRCR domains have been directly implicated in ligand binding. These include CD6 signalling via CD166, CD163-mediated clearance of the haemoglobin–haptoglobin complex, Mac-2 binding protein's (M2bp's) interaction with matrix components, and the binding of microbial ligands by the scavenger receptors SR-A1, SPα, and MARCO (22, 23, 24, 25, 26, 27). Although the multiple SALSA SRCR domains likewise have been implicated in ligand binding, the molecular basis for its diverse interactions remains unknown.

To understand the multiple ligand-binding properties of the SALSA molecule, we undertook an X-ray crystallographic study to provide detailed information of the SALSA interaction surfaces. We here provide the atomic resolution structures of SALSA SRCR domains 1 and 8. We identify cation-binding sites and demonstrate their importance for ligand binding. By comparing our data to previously published structures of SRCR domains, we propose a

[1]Sir William Dunn School of Pathology, University of Oxford, Oxford, UK   [2]Institute of Dentistry, University of Turku, Turku, Finland   [3]Central Oxford Structural Molecular Imaging Centre, University of Oxford, Oxford, UK

Correspondence: steven.johnson@path.ox.ac.uk; martinpreichhardt@gmail.com

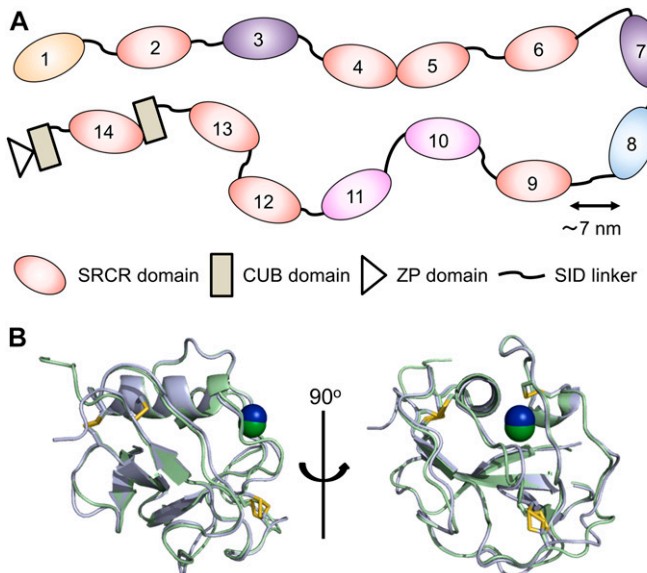

**Figure 1. Crystal structure of SALSA domains SRCR1 and SRCR8.**
**(A)** Schematic representation of the domain organization of full-length SALSA. SRCR1 and SRCR8 are highlighted in green and blue, respectively. All SRCR domains share >88% sequence identity. 100% identity is shared by SRCR3 and 7 (yellow) and SRCR10 and 11 (purple). **(B)** Front and side views of an overlay of SRCR1 (green) and SRCR8 (blue), showing four conserved disulphide bridges (yellow). Both SRCR1 and SRCR8 were found to coordinate a metal ion, modelled as $Mg^{2+}$ (dark green for SRCR1 and dark blue for SRCR8). The limited structural variation observed between SRCR1 and SRCR8 (92% sequence identity) imply that these are appropriate representations of all SALSA SRCR domains 1–13.

generalised binding mechanism for this ancient, evolutionarily conserved, fold.

## Results

The scavenger receptor SALSA has a very wide range of described ligands, including microbial, host innate immune, and ECM molecules. To understand the very broad ligand-binding abilities of the SALSA molecule, we applied a crystallographic approach to determine the structure of the ligand-binding SRCR domains of SALSA. SRCR domains 1 and 8 (SRCR1 and SRCR8) were expressed in *Drosophila melanogaster* Schneider S2 cells with a C-terminal His-tag. The domains were purified by Ni-chelate and size-exclusion chromatography (SEC) and crystallised, and the structures were solved by molecular replacement. This yielded the structures of SRCR1 and SRCR8 at 1.77 and 1.29 Å, respectively (Fig 1B). (For crystallographic details, see Table 1).

The SALSA SRCR domains reveal a classic globular SRCR-fold, with four conserved disulphide bridges, as described for the SRCR type B domains. The fold contains one α-helix and one additional single helical turn. The N and C termini come together in a four-stranded β-sheet. SALSA SRCR1-13 are highly conserved, with 88–100% identity. Variation is only observed in 9 of the 109 aa residues, all of these observed in peripheral loops, without apparent structural significance. Combined, the data from SRCR1 and SRCR8 are thus valid representations of all SALSA SRCR domains. Both SRCR1 and SRCR8

are stabilized by a metal ion buried in the globular fold. The placement suggests the ion is bound during the folding of the domain and is modelled as $Mg^{2+}$, which is present in the original expression medium and in the crystallisation conditions of SRCR1.

So far, all described ligand-binding interactions of SALSA have been shown to be $Ca^{2+}$-dependent. We therefore proceeded to address the ligand-binding potential of the SRCR domains by adding $Ca^{2+}$, $Mg^{2+}$, and a cocktail of sugars to SRCR8 crystals before freezing. This yielded a second crystal form of SRCR8 with the original $Mg^{2+}$ ion, site 1, and two additional cations bound, sites 2 and 3 (Fig 2). All three sites are class three cation-binding sites, with the coordination obtained from residues in distant parts of the sequence (28).

Assignment of the identity of the ions at the paired site was carried out by modelling $Mg^{2+}$ or $Ca^{2+}$ at each site, followed by refinement of the structure and analysis of the difference maps (Fig S1). These revealed that $Mg^{2+}$ best satisfied the data at both sites, consistent with the 20-fold molar excess of $Mg^{2+}$ over $Ca^{2+}$ in the crystallisation solution. However, it is worth noting that either site could likely accommodate either cation depending on local concentration. Analysis of the bond lengths and coordination numbers suggest that site 2 is a canonical $Mg^{2+}$ site, with octahedral geometry and average bond lengths of 2.1 Å, while site 3 displays a higher coordination number and longer bond lengths, more consistent with a $Ca^{2+}$-binding site (29). The $Mg^{2+}$ at site 1 is coordinated by the backbone carbonyl groups of S1021 and V1060, as well as the side chains of D1023 and D1026, and two waters, and is buried in the domain fold (Fig 2C). The $Mg^{2+}$ at site 2 is coordinated by D1019, D1020, and E1086 plus three waters (Fig 2D). The $Mg^{2+}$ at site 3 is coordinated by the side chains of D1020, D1058, D1059, and N1081, with additional contributions from a water and an extra density (Fig 2E). Attempts to model this extra density as any of the sugars or alcohols present in the crystallisation solution failed to produce a satisfactory fit; therefore, it likely represents a superposition of a number of molecules.

In contrast to the $Mg^{2+}$ at site 1, these cations at sites 2 and 3 are exposed on the surface of the domain, and the protein only contributes a fraction of the coordination sphere, with the remainder contributed by waters or small molecules from the crystallisation solution. According to the literature, the majority of described SALSA ligands are negatively charged. Thus, the surface-exposed cations likely provide a mechanism for ligand binding for the SALSA SRCR domains, whereby the anions of the ligand substitute for the waters or the density at site 3 observed in our structure. To test this hypothesis, site-directed mutagenesis was used, targeting the key residues coordinating sites 2 and 3. Included in the further analysis were single mutations D1019A and D1020A. While mutation of D1019 is expected to only disrupt binding of cations at site 2, mutation of the shared D1020, will likely affect binding of both cations.

As SALSA recognizes a very broad range of biological ligands, we set out to test the effect of SRCR domain point mutations on interactions with a wide array of biological ligands. These included binding to (1) hydroxyapatite, a phosphate-rich mineral essential for the binding of SALSA to the teeth surface, where it mediates antimicrobial effects (30); (2) heparin, a sulphated glycosaminoglycan as a mimic for the ECM/cell surface, for which binding of SALSA has been described to affect cellular differentiation and microbial colonisation (31); (3) Group A *Streptococcus* surface protein, Spy0843, a leucine-rich repeat protein demonstrated to bind to SALSA (32) (Fig 3).

**Table 1. Data collection and refinement statistics (molecular replacement).**

| | SRCR1 (pdbid: 6sa4) | SRCR8 (pdbid: 6sa5) | SRCR8soak (pdbid: 6san) |
|---|---|---|---|
| Data collection | | | |
| Space group | P 2$_1$ 2$_1$ 2$_1$ | P 2$_1$ 2$_1$ 2$_1$ | P 1 2$_1$ 1 |
| Cell dimensions | | | |
| $a, b, c$ (Å) | 36.77, 45.19, 69.37 | 32.82, 40.82, 62.99 | 27.24, 46.64, 93.63 |
| $\alpha, \beta, \gamma$ (°) | 90.00, 90.00, 90.00 | 90.00, 90.00, 90.00 | 90.00, 97.37, 90.00 |
| Resolution (Å) | 28.52–1.77 (1.80–1.77)[a] | 40.82–1.29 (1.31–1.29) | 46.64–1.36 (1.39–1.36) |
| $R_{merge}$ | 0.17 (1.36) | 0.117 (1.12) | 0.074 (0.801) |
| $I/\sigma I$ | 8.1 (1.1) | 8.6 (0.9) | 14.9 (2.2) |
| Completeness (%) | 99.8 (99.3) | 100 (99.6) | 98.3 (96.9) |
| Redundancy | 6.3 (6.6) | 11.4 (8.0) | 6.6 (6.4) |
| Refinement | | | |
| Resolution (Å) | 28.52–1.77 (1.95–1.77) | 34.27–1.29 (1.35–1.29) | 30.95–1.36 (1.39–1.36) |
| No. of reflections | 11,730 | 21,958 | 49,166 |
| $R_{work}/R_{free}$ | 0.186/0.229 (0.266/0.348) | 0.155/0.188 (0.326/0.279) | 0.186/0.226 (0.267/0.326) |
| No. of atoms | | | |
| Protein | 824 | 829 | 3,192 |
| Ligand/ion | 26 | 18 | 34 |
| Water | 109 | 124 | 358 |
| $B$-factors | | | |
| Protein | 22.36 | 16.80 | 17.48 |
| Ligand/ion | 49.33 | 54.03 | 26.08 |
| Water | 31.15 | 33.66 | 35.55 |
| R.m.s. deviations | | | |
| Bond lengths (Å) | 0.006 | 0.009 | 0.007 |
| Bond angles (°) | 0.809 | 1.026 | 0.87 |
| Ramachandran outliers | 0 | 0 | 0 |
| Rotamer outliers | 0 | 0 | 0 |

Number of crystals was one for each structure.
[a]Values in parentheses are for highest resolution shell. Data from SRCR1 and SRCR8 crystals were collected on Diamond beamline I04, while data for SRCR8soak were collected on beamline I03.

Different binding assays provide an understanding of a generalised binding mechanism of SALSA SRCR domains. While the WT SRCR domain bound to all three ligands, both of the cation-binding site mutations, D1019A and D1020A, abolished binding. This is consistent with the bound cations acting as a bridge for ligand interaction and thus provides a mechanistic explanation for the binding properties of the SALSA SRCR domains. In the literature, SALSA ligand binding has been described as specifically calcium dependent. To verify this, we conducted binding assays in an MgEGTA-containing buffer (Fig 3C). The exchange of magnesium for calcium abolished ligand interactions, thus supporting a calcium-specific mediation of binding. As mutation of site 2 alone (modelled as Mg$^{2+}$ in our structure) abolished binding, our data suggest that Ca$^{2+}$ may occupy site 2 under physiological conditions.

All known members of the SRCR superfamily share a very high degree of identity, both at the sequence and structural levels. An FFAS search (33) of the SRCR8 sequence showed highest similarities

to CD163 SRCR5 (score: −65.4, 46% identity), M2bp (score: −64.4, 54% identity), neurotrypsin (score: −61.5, 50% identity), MARCO (score: −59.4, 50% identity), CD5 SRCR1 (score: −48.8, 26% identity), and CD6 SRCR2 (score: −45.6, 60% identity). Using the Dali server (34), searches for the cation-binding SRCR8 soak structure identified two top hits as M2bp (pdbid: 1by2) and CD6 SRCR3 (pdbid: 5a2e). These were identified with respective Z-scores of 21.4 (r.m.s.d. of 1.1 Å with 106 of 112 residues aligned) and 20.4 (r.m.s.d. of 1.5 Å with 109 of 109 residues aligned). Despite the classical division of SRCR superfamily proteins into groups A and B, based on the conserved three versus four cysteine bridges, the SRCR fold is very highly conserved, and the SALSA SRCR domain structures correlate closely to both group A and group B SRCR superfamily domains (Fig 4A).

For members of the SRCR superfamily where the SRCR domain directly partakes in ligand binding, both microbial and endogenous protein ligands have been described. For MARCO, crystallographic structures identified a cation-binding site exactly corresponding to

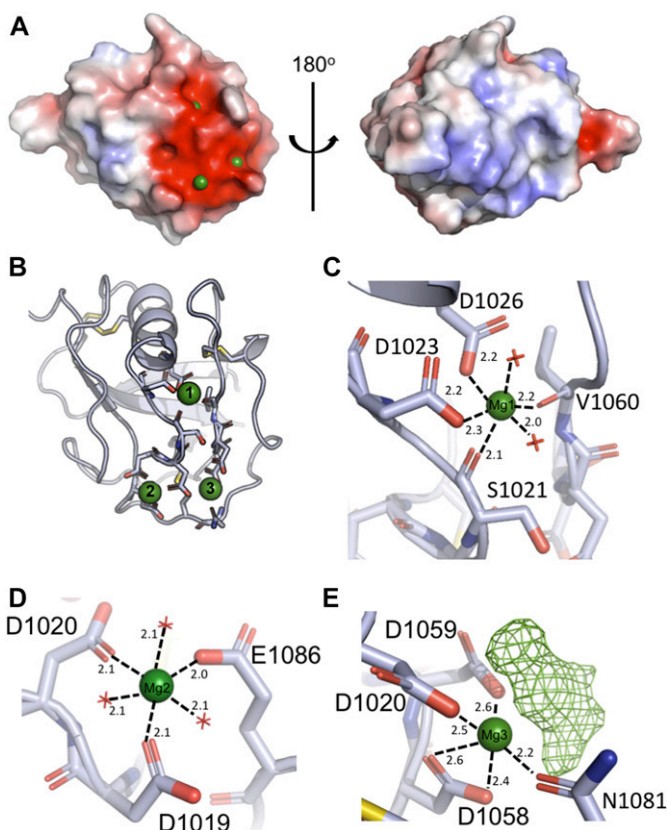

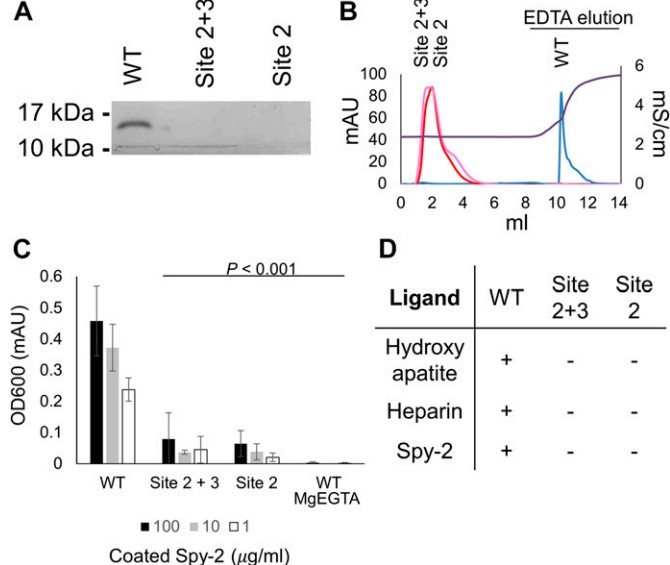

**Figure 2. Crystal structure of SRCR8 with bound magnesium ions indicates mechanism of ligand binding.**
**(A)** Surface charge distribution of SRCR8 (calculated without the presence of cations) shows a positive cluster on one side with a strong negative cluster on the other. The negative cluster expands across ~300 Å² and mediates the binding of three cations (green). **(B)** Representation of the residues coordinating the three cations. The upper $Mg^{2+}$, site 1, sits somewhat buried in the structure and may be essential for structural stability. The lower cations at sites 2 and 3 are more exposed. **(C)** Detailed view of the coordination of the upper $Mg^{2+}$, site 1. The coordination number of six is achieved by two waters, two backbone carbonyls, and two side chain carboxylates. **(D)** Detailed view of the coordination of the cation at site 2, modelled as $Mg^{2+}$ based on bond length and coordination number. Here, the coordination number of six is achieved by three waters and three side chain carboxylates. **(E)** Detailed view of the coordination of the cation at site 3, modelled as $Mg^{2+}$. The coordination is achieved by three side chain carboxylates, one side chain amide, and an unmodelled density that is assumed to be a superposition of crystallisation condition compounds.

**Figure 3. Mutating the cation-binding residues of scavenger receptor cysteine-rich (SRCR) domains abolish function.**
Through multiple ligand-binding assays, we demonstrated the functional importance of cation binding by the SRCR domains. Mutations affecting site 2 (D1019A) and mutations affecting sites 2 and 3 (D1020A) both abolish function. **(A)** WT and mutant forms of SRCR8 were incubated with hydroxyapatite beads in a $Ca^{2+}$-containing buffer. After extensive washing, bound protein was eluted with EDTA. Eluted fractions were run on a 4–20% SDS–PAGE gel and visualized by Coomassie staining. Only WT SRCR8 bound hydroxyapatite. **(B)** WT and mutant forms of SRCR8 were flown over a heparin (HiTrap HP, 1 ml) column in a $Ca^{2+}$-containing buffer. Protein bound to the column was eluted with 0.5 M EDTA. Only WT SRCR8 bound the heparin column. Traces: SRCR8 (blue), D1020A (pink), D1019A (red), conductivity (brown). **(C)** In an ELISA-based setup, a concentration range of the Spy-2 domain of Spy0843 was coated (1–100 µg/ml). WT and mutant SRCR8 domains were added (100 µg/ml), and binding was detected with a monoclonal anti-SALSA antibody. Binding was only observed for WT SRCR8. **(D)** Overview of ligand-binding studies; + denotes binding, – denotes no binding.

site 2 in the SALSA SRCR8 domain (26) and point mutations of this site abolished function. Common to the MARCO and SALSA SRCR domains is the cluster of negatively charged residues coordinating the functionally important cations. A similar cluster is also observed in the SRCR3 domain of CD6 and has been shown by mutagenesis to be directly involved in binding to the human surface receptor CD166. Indeed, a point mutation of D291A (corresponding to D1019 of SALSA-SRCR8) reduced the ligand-binding potential of CD6 to less than 10% (27). Furthermore, mutational studies of SRCR domains 2 and 3 from CD163 proved an involvement of this specific site in the binding of the haemoglobin–haptoglobin complex (36). All structural evidence from mutational studies of SRCR domains thus indicate a conserved surface-mediating ligand binding (Fig 4B).

Interestingly, various levels of calcium-dependency on ligand interactions have been described for all SRCR domains directly involved with binding. SR-A1, Spα, MARCO, CD5, and CD6 all rely on calcium for interactions with microbial ligands (24, 25, 26, 37, 38, 39). Furthermore, the binding of CD163 to the haemoglobin–haptoglobin complex is calcium-dependent, while CD6 also recognizes endogenous surface structures (other than CD166) in a calcium-dependent manner (40). This suggests that the cation-dependent binding mechanism identified for SALSA is a general conserved feature of all SRCR domains. Indeed, sequence alignment of SRCR domains from 10 different SRCR superfamily proteins, all with SRCR domains directly involved with ligand binding, reveals a very high level of conservation of the two cation-binding sites identified in the SALSA domains (Fig 4C). The ConSurf server is a tool to estimate (on a scale from 1 to 9) the level of evolutionary conservation of residues in a given fold (41). A search with the SRCR8 model shows that D1019, D1020, D1058, and E1086 all score 7 (highly conserved), while N1081 and D1059 score 6 and 4, respectively (thus less conserved). Whenever sequence identity is not conserved, substitutions are observed with other residues overrepresented in cation-binding sites (D, E, Q, and N) (28, 29). The cation-binding sites identified in the SALSA SRCR domains, thus, appear to be a highly conserved feature of the general SRCR fold.

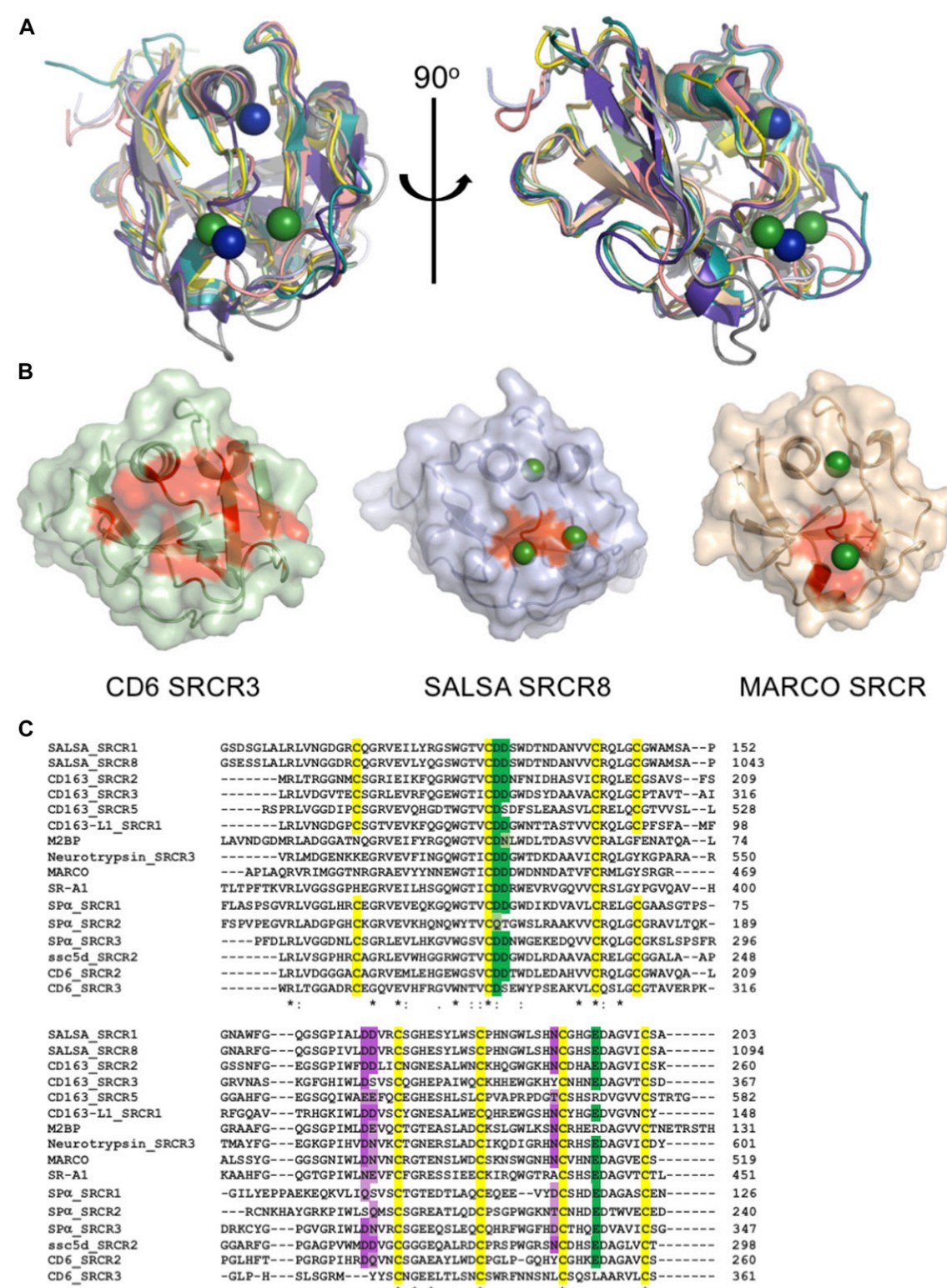

**Figure 4. Conserved ligand-binding motif across scavenger receptor cysteine-rich (SRCR) domains.**
SRCR domains from multiple proteins engage in cation-dependent ligand interactions. **(A)** Structural overlay of domains from seven SRCR superfamily proteins, all with ligand binding mediated through the SRCR domain. This reveals a highly conserved fold across both type A and type B SRCR domains. SRCR1 (pale green), SRCR8 (light blue), MARCO (pdbid: 2oy3, sand), CD163 (pdbid: 5jfb, purple), CD5 (pdbid: 2OTT, grey), CD6 (pdbid: 5a2e, pink), M2bp (pdbid: 1by2, yellow), and murine neurotrypsin (pdbid: 6h8m, teal) (35). SALSA magnesium: green, MARCO magnesium: blue. **(B)** Surface representation of CD6 SRCR3, SALSA SRCR8, and MARCO in same orientation. Point mutations with a verified impact on ligand binding are highlighted in red, indicating a conserved surface involved in ligand binding. Bound magnesium is highlighted in

Structural inspection of the overlay of SALSA SRCR8 with the corresponding region in the other known SRCR domain structures clearly shows the potential for cation binding at these sites. We thus propose that the cation-binding sites identified here are an essential feature of the ancient SRCR fold and are a conserved mechanism responsible for mediating ligand binding in the class of SRCR superfamily proteins.

# Discussion

Although SALSA has previously been described to interact with a wide range of biological ligands, little has been known of the binding mechanisms. Furthermore, it has not been known if SALSA interacts with various ligands in a similar way or if distinct binding sites are used. Here, we demonstrate that mutations of a dual cation-binding site interrupt interactions with a representative selection of very different types of ligands. Specific disruption of site 2 was sufficient to abolish ligand binding. We modelled the cation at site 2 in our crystal as $Mg^{2+}$, based on an analysis of bond length, coordination number, and behaviour of crystallographic refinements with different cations modelled. However, experimental data demonstrated that binding to the ligands tested was only dependent on the presence of $Ca^{2+}$ and not $Mg^{2+}$. This is in line with previous descriptions of most SALSA–ligand interactions (5, 6, 42). In the extracellular compartment, the molar concentration of $Ca^{2+}$ is higher than $Mg^{2+}$ (2.5 and 1 mM for calcium and magnesium, respectively), and it is therefore likely that both sites 2 and 3 will be occupied by $Ca^{2+}$ in a physiological setting. The identification of this dual $Ca^{2+}$-binding site thus provides an explanation for the $Ca^{2+}$ dependency of all SALSA–ligand interactions described in the literature, suggesting this mechanism of binding is applicable to all SALSA SRCR ligands. Multiple studies have proposed a role for the motif GRVEVxxxxxW in ligand binding (43, 44, 45). The crystal data show that this peptide sequence is buried in the SRCR fold, and we found no validation for a role in ligand binding, although mutations within this sequence are likely to perturb the overall fold. This motif thus does not appear to have any physiological relevance as defining a ligand-binding site.

The conserved usage of a single ligand-binding area for multiple interactions suggests that each SALSA SRCR domain engages in one ligand interaction. A common feature of the ligands described here, as well as a number of other ligands such as DNA and LPS, is the presence of repetitive negatively charged motifs (31). We analysed ligand binding by individual SRCR domains in surface-plasmon resonance and isothermal calorimetry assays, but interactions were observed to be of very low affinity, making reliable measurements unfeasible. This is not surprising for a molecule such as SALSA, where the molecular makeup with the full extension of 13 repeated units, interspersed by predicted nonstructured flexible SIDs, provides a molecule that can generate high-avidity interactions with repetitive ligands, despite having only low-affinity interactions for an individual domain. Furthermore, it has been suggested that SALSA in body secretions may oligomerize into larger complexes (5, 46, 47, 48), probably via the C-terminal CUB and zona pellucida domains. The

repetitive nature and possible oligomerization allow SALSA to not only engage with a repetitive ligand on one surface (e.g., LPS or Spy0843 on microbes) but also engage in multiple ligand interactions simultaneously. This would be relevant for its interactions with other endogenous defence molecules, such as IgA, SPs, and complement components, where a cooperative effect on microbial clearance has been demonstrated (12, 16, 17, 49). In addition, this model of multiple ligand binding would be relevant for microbes described to use SALSA for colonisation of the teeth or the host epithelium (10, 50, 51) (Fig 5).

SALSA belongs to the SRCR superfamily, a family of proteins characterised by the presence of one or more copies of the ancient and evolutionarily highly conserved SRCR fold (52). Although a couple of SRCR domains, such as the ones found in complement factor I and hepsin, have not been described to bind ligands directly, most others have (53, 54). SRCR superfamily members, such as SALSA, SR-A1, Spα, SSc5D, MARCO, CD6, and CD163, have broad scavenger-receptor functions, recognizing a broad range of microbial surface structures and mediate clearance (24, 25, 26, 37). Although this potentially is relevant for all SRCR superfamily proteins, some members of the family have distinct protein ligands, such as CD6, CD163, and M2bp (22, 23, 27, 55). With the exception of the CD6–CD166 interactions, most described SRCR–ligand interactions are calcium dependent, irrespective of the ligand (24, 25, 26, 37, 38, 39, 40). A cation-binding site is conserved across SRCR domains, and multiple studies support a role for this site in ligand binding. Even the specialised CD6–CD166 interaction uses the same surface for binding, despite "having lost" the calcium dependency (27).

Our studies have thus identified a dual cation-binding site as essential for SALSA–ligand interactions. Analysis of SRCR folds from various ligand-binding domains reveals a very high level of conservation of the residues at this dual site. The conservation of this site, along with the well-described cation dependency on most SRCR–ligand interactions, suggests that the binding mechanism described for the SALSA SRCR domains is applicable to all SRCR domains. We thus propose to have identified in SALSA a conserved functional mechanism for the SRCR class of proteins. This notion is further supported by the specific lack of conservation of these residues observed in the SRCR domains of complement factor I and hepsin, where no ligand binding has been shown. The SRCR domains in these two molecules may thus represent an evolutionary diversion from the common broad ligand-binding potential of the SRCR fold. The novel understanding of the SRCR domain generated here will allow for an interesting future targeting of other SRCR superfamily proteins, with the potential of modifying function.

# Materials and Methods

## Expression of recombinant proteins

### Insect cell expression
Codon-optimized DNA (GeneArt; Thermo Fisher Scientific) was cloned into a modified pExpreS2-2 vector (ExpreS2ion Biotechnologies) with a

green. (C) Clustal Omega (EMBL-EBI) sequence alignment of SRCR domains from 10 SRCR superfamily proteins. Conservation of the cation-binding sites are displayed in green (site 2) and purple (site 3). Dark colouring indicates 100% identity with the SALSA sites, and lighter colouring indicates conservation of residues commonly implicated in cation-binding (D, E, Q, or N). Cysteines are highlighted in yellow, and overall sequence identity is denoted by * (100%), : (strongly similar chemical properties), and . (weakly similar chemical properties).

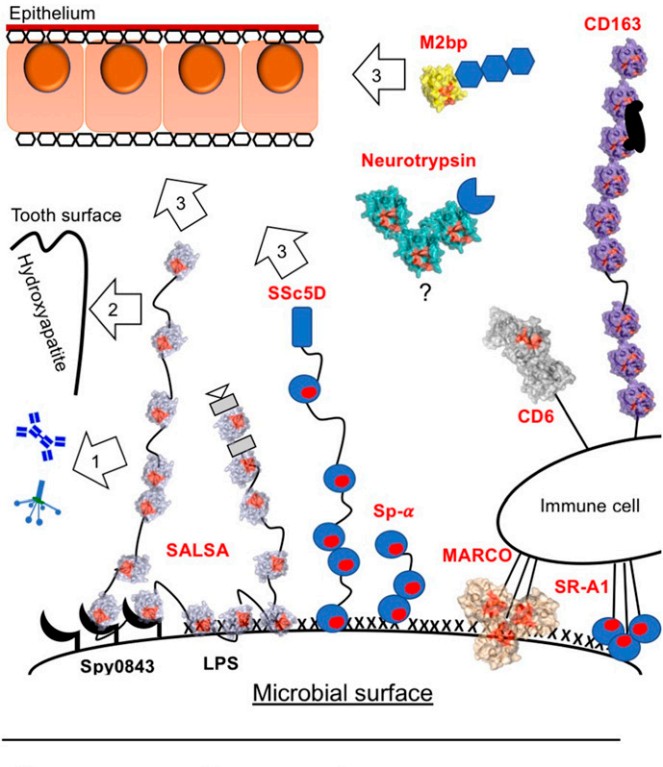

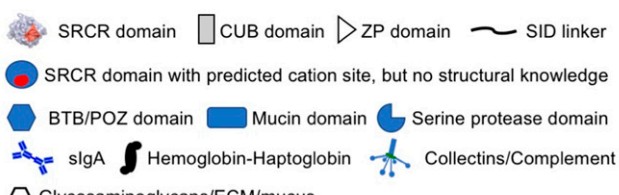

**Figure 5. SALSA scavenger receptor cysteine-rich (SRCR) cation-binding motif reveals a conserved mechanism for broad-spectrum ligand interactions of SRCR superfamily molecules.**

Based on mutational studies and structural information across SRCR proteins, we propose a generalised mechanism of ligand interaction mediated by the cation-binding surface motif of the evolutionarily ancient SRCR fold (left side). SALSA has been described to bind a broad range of ligands, incorporating into a complex network of binding partners on the body surfaces and the colonizing microbiota. The multiple SRCR domains of full-length SALSA bind repetitive targets (both protein and carbohydrate structures) on the surface of microbes. The secreted fluid-phase molecule may thus lead to microbial agglutination and clearance. However, the repetitive form of binding sites will allow for simultaneous binding to endogenous targets as well. This being, for example, 1) binding of IgA, collectins, and complement components to induce a cooperative antimicrobial effect; 2) binding of hydroxyapatite on the tooth surface; and 3) ECM proteins and glycosaminoglycans, as well as mucus components of the epithelial surface (such as heparin, galectin 3, and mucins) (right side). The cation-binding motif described in SALSA is conserved in most other SRCR proteins. For CD6, CD163, and MARCO, mutational studies support a crucial role for this area in ligand interactions. CD163 binds the haemoglobin–haptoglobin complex and microbial surfaces. CD6 binds endogenous ligands but also engages in microbial binding. MARCO forms multimers and binds microbial surface structures. Other SRCR proteins with similar functions and conserved cation sites include SR-A1, Sp-α, SSc5D, and M2bp. The functional role of the neurotrypsin SRCR domains is not known. The remarkable repetitive formation of multiple SRCR domains in many SRCR superfamily proteins, with several domains containing a binding site with a broad specificity, would supposedly allow for interactions with multiple ligands simultaneously. The SRCR fold thus appears to be an important functional component of scavenging molecules engaging in complex network of interactions. The multiple SRCR domains shown for SALSA, CD163, and

C-terminal His-6 tag. The purified plasmid was transformed into S2 cells grown in EX-CELL 420 (Sigma-Aldrich) with 25 µl ExpreS2 Insect-TR 5X (ExpreS2ion Biotechnologies). Selection for stable cell lines (4 mg/ml geneticin [Thermo Fisher Scientific]) and expansion were carried out according to the manufacturer's instructions.

### Escherichia coli *expression*

DNA strings (GeneArt; Thermo Fisher Scientific) were cloned into pETM-14 and transformed into M15pRep cells. Protein expression was carried out in LB media (with 30 µg/ml kanamycin). Cells were induced with 1 mM IPTG. The cultures were centrifuged (3,220$g$, 15 min) and the cell pellets resuspended and lysed in PBS containing 1 mg/ml DNase and 1 mg/ml lysozyme.

### Protein purification

#### *SRCR domains*

Insect culture supernatant was collected by centrifugation (1,000$g$ at 30 min), filtered and loaded onto a Roche cOmplete Ni$^{2+}$-chromatography column (1 ml, Cat. no. 06781543001; Sigma-Aldrich), washed in 20 CV buffer (50 mM Tris, pH: 9.0, 200 mM NaCl). Bound protein was eluted with 250 mM imidazole. Following this, SEC was carried out on a Superdex 75 16/60 HR column (GE Healthcare) equilibrated in 10 mM Tris, pH: 7.5, 200 mM NaCl.

#### *Spy-2*

Lysed cell pellets were homogenized and centrifuged at 20,000$g$ for 30 min. The filtered supernatant was loaded onto a Ni$^{2+}$-chromatography column (5 ml; QIAGEN) and washed in 20 CV buffer (50 mM Tris, pH: 8.5, 200 mM NaCl, 20 mM imidazole). Bound protein was eluted (in 50 mM Tris, pH: 8.5, 200 mM NaCl, 250 mM imidazole), concentrated, and subjected to SEC (Superdex 75 16/60 HR column; GE Healthcare).

### Crystallisation, X-ray data collection, and structure determination

Purified SRCR1 and SRCR8 were concentrated to 20 mg/ml. SRCR1 was mixed with an equal volume of mother liquor containing 0.2 M MgCl$_2$ hexahydrate, 10% (wt/vol) PEG8000, 0.1 M Tris, pH: 7.0, and crystallised in 400 nl drops by the vapor diffusion method at 21°C. SRCR8 was mixed with an equal volume of mother liquor containing 0.1 M LiSO$_4$, 20% (wt/vol) PEG6000, 0.01 M Hepes, pH: 6.5, and crystallised in 800 nl drops. For SRCR8 + cation crystals were grown in 0.2 M MgCl$_2$ hexahydrate, 20% (vol/vol) isopropanol, 0.1 M Hepes, pH: 7.5, and crystallised in 400 nl drops. The crystallisation buffer was supplemented with 10 mM Mg$^{2+}$ and 10 mM Ca$^{2+}$, as well as 10 mM maltose, D-galactose, D-saccharose, D-mannose, D-glucose, and sucrose octasulphate (all Sigma-Aldrich), 24 h prior to freezing. All crystals were cryoprotected in mother liquor supplemented with 30% glycerol and flash frozen in liquid N$_2$. Data were collected at a temperature of 80 K

neurotrypsin are represented as copies of protein-specific SRCR domains with known structure. Conserved cation-coordinating residues are highlighted in red. SRCR8soak (light blue), MARCO (pdbid: 2oy3, sand), CD163 (pdbid: 5jfb, purple), CD6 (pdbid: 5a2e, grey), M2bp (pdbid: 1by2, yellow), and murine neurotrypsin (pdbid: 6h8m, teal).

on beamlines I04, at a wavelength of 1.0718 Å (for SRCR1 and SRCR8) and I03, at a wavelength of 0.9762 Å (for SRCR8cat) at the Diamond Light Source, as specified in Table 1. The structure of SRCR8 was solved by molecular replacement using MolRep within CCP4 (56) with the structure of CD6 SRCR domain 3 (PDB ID 5a2e (27)). The structures of SRCR1 and SRCR8 soaked in cations were solved by molecular replacement using the structure of SRCR8. Refinement and re-building were carried out in Phenix and Coot (57, 58). Assignment of metal ions was carried out by first refining the structure without anything in the metal binding sites, followed by addition of combinations of probable ligands, and re-refinement in phenix.refine using restraints generated by phenix.ready_set for each combination. The structures were characterised by the statistics shown in Table 1 with no Ramachandran outliers. Protein structure figures were prepared using Pymol version 2.0 (Schrödinger, LLC).

## Hydroxyapatite binding assay

150 $\mu$l hydroxyapatite nanoparticle suspension (Cat. no. 702153; Sigma-Aldrich) was washed into buffer (10 mM Hepes, pH: 7.5, 150 mM NaCl, 1 mM $Ca^{2+}$). Beads were incubated in 80 $\mu$l SRCR8, SRCR8 D34A, or SRCR8 D35A (all at 0.5 mg/ml in the same buffer) with shaking for 1 h at RT. Beads were spun and washed 6× in 1 ml buffer. Bound protein was eluted in 100 $\mu$l 0.5 M EDTA and visualized by SDS–PAGE (4–20%; Bio-Rad) and Coomassie staining (Instant Blue, Expedeon).

## Heparin binding assay

SRCR8, SRCR8 D34A, or SRCR8 D35A in 10 mM Hepes, pH: 7.5, 10 mM NaCl, 1 mM $Ca^{2+}$ were loaded onto a HiTrap Heparin HP column (1 ml; GE Healthcare), equilibrated in the same buffer. Bound protein was then eluted with 10 mM Hepes, pH: 7.5, 10 mM NaCl, 20 mM EDTA.

## Spy-2 binding assay

On a MaxiSorp plate (Nunc), 100 $\mu$l purified Spy-2 was coated O/N at 4°C in a concentration ranging from 0.032 to 3.2 $\mu$M in coating buffer (100 mM $NaHCO_3$ buffer, pH: 9.5). The plate was blocked in 1% gelatine in PBS, and SRCR8, SRCR8 D34A, and SRCR8 D35A were added (all at 7.1 $\mu$M in 10 mM Hepes, pH 7.5, 150 mM NaCl, 1 mM $Ca^{2+}$, 0.05% Tween20). Bound protein was detected with monoclonal anti-SALSA antibody diluted 1:10,000 (1G4; Novus Biologicals) and HRP-conjugated rabbit anti-mouse antibody 1:10,000 (W4028; Promega). The plate was developed with 2,2′-azino-bis(3-ethylbenzothiazoline-6-sulfonic acid) (Sigma-Aldrich) and analysed by spectrophotometry at 405 nm. To test calcium-specific dependency of the interaction, the WT assay above was repeated in a buffer containing 10 mM Hepes, pH 7.5, 150 mM NaCl, 1 mM $Mg^{2+}$, 1 mM EGTA, and 0.05% Tween20.

## Data availability

Structure factors and coordinates from this publication have been deposited to the PDB database https://www.wwpdb.org and assigned the identifiers: SRCR1 pdbid: 6sa4, SRCR8 pdbid: 6sa5, SRCR8soak with three cations pdbid: 6san.

# Supplementary Information

# Acknowledgements

We acknowledge Diamond Light Source and the staff of beamlines I03 and I04 for access under proposal MX18069. The Central Oxford Structural Molecular and Imaging Centre is supported by the Wellcome Trust (201536). MP Reichhardt was financially supported by grants from the Wihuri Foundation and the Finnish Cultural Foundation. Staff and experimental costs in SM Lea laboratory were supported by a Wellcome Investigator Award (100298) and an Medical Research Council (UK) programme grant (M011984). V Loimaranta was supported by the Turku University Foundation.

## Author Contributions

MP Reichhardt: conceptualization, data curation, formal analysis, funding acquisition, investigation, and writing—original draft, review, and editing.
V Loimaranta: resources.
SM Lea: conceptualization, data curation, formal analysis, funding acquisition, validation, investigation, methodology, project administration, and writing—review and editing.
S Johnson: conceptualization, data curation, formal analysis, supervision, validation, investigation, visualization, methodology, project administration, and writing—original draft, review, and editing.

## Conflict of Interest Statement

The authors declare that they have no conflict of interest.

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
