## [Reviewer comments · Life Science Alliance]

Life Science Alliance

Structures of SALSA domains reveal the conserved ligand-binding mechanism of the ancient SRCR-fold

Martin Reichhardt, Vuokko Loimaranta, Susan Lea, and Steven Johnson

DOI: <https://doi.org/10.26508/lsa.201900502>

Corresponding author(s): Martin Reichhardt, University of Oxford and Steven Johnson, University of Oxford

Review Timeline:

Submission Date:	2019-07-26
Editorial Decision:	2019-08-15
Revision Received:	2020-01-19
Editorial Decision:	2020-02-06
Revision Received:	2020-02-14
Accepted:	2020-02-14

Scientific Editor: Andrea Leibfried

Transaction Report:

August 15, 2019

Re: Life Science Alliance manuscript #LSA-2019-00502-T

Prof. Susan M Lea
Oxford, University of
Sir William Dunn School of Pathology
South Parks Raod
South Parks Road
Oxford, UK-Oxford OX1 3RE
United Kingdom

Dear Dr. Lea,

Thank you for submitting your manuscript entitled "The structure of SALSA/DMBT1 SRCR domains reveal the conserved ligand-binding mechanism of the ancient SRCR-fold" to Life Science Alliance. The manuscript was assessed by expert reviewers, whose comments are appended to this letter.

As you will see, reviewer #1 is essentially happy as is, but reviewer #2 thinks that further support for your conclusions is needed. The reviewer provides constructive input on how to provide such support, and we would thus like to invite you to submit a revised version of your manuscript to us. The additional lines of support as requested by reviewer #2 seem straightforward to provide, but please get in touch in case you would like to discuss individual revision points further with us.

Thank you for this interesting contribution to Life Science Alliance. We are looking forward to receiving your revised manuscript.

Sincerely,

B. MANUSCRIPT ORGANIZATION AND FORMATTING:

Reviewer #1 (Comments to the Authors (Required)):

The manuscript of Reichhardt et al. describes the structure determination of the SRSC domains 1 and 8 from SALSA/DMBT1 using X-ray crystallography. This protein is involved in recognition of different ligands relevant to mucosal immunology. Assays were carried out to test binding of structurally unrelated ligands, and mutation of residues involved in single and dual binding of

calcium atoms (D1019A and D1020A) showed that Ca^{2+} mediates ligand binding.

As pointed out by the authors, these domains have several homologues of known structure and high sequence similarity. Also, the role of structurally equivalent residues involved in calcium binding has been tested previously in these homologues by mutagenesis and binding assays.

I appreciate that the authors carried out SPR binding assays to obtain binding constants of individual domains but observed only weak interactions, and to me the main novelty of the manuscript is in Fig. 3, showing experiments to test binding against hydroxyapatite, heparin and the Spy-2 domain of Spy0843. Comparison of the binding data of WT and mutant domains demonstrates that calcium is essential for a unified mode of ligand binding, an important point to make considering that other SRSC domains can bind ligands using other regions on the domain surface.

In page 14 of the discussion, lines 9-11 it says "... experimental data demonstrated that binding to the ligands tested was only dependent on the presence of Ca^{2+} , and not Mg^{2+} ". To test this, binding assays of the WT SRSC8 domain were carried out in 1mM Ca^{2+} and also in 1mM EGTA (chelator that binds Ca^{2+} with greater affinity than Mg^{2+}) and 1mM Mg^{2+} . Strictly, I am not convinced that the domain would not bind ligands using Mg^{2+} if that was the only metal available, but their argument that Ca^{2+} levels in the extracellular medium are much higher than those of Mg^{2+} does suggest that Ca^{2+} is more biologically relevant.

The crystallographic data and figures are clearly laid out.
The paper is carefully written and I could not spot any typos.

I recommend publication of the manuscript without any further changes.

Reviewer #2 (Comments to the Authors (Required)):

The manuscript by Reichardt et al describes crystal structures of two SRCR domains from SALSA. Based on mutational analysis, in vitro interaction experiments and bioinformatics they propose that a binding site for a divalent metal ion is conserved in many other SRCRs and that they mediate ligand binding. Their results are potentially of significant interest to investigators working with SRCR containing proteins. However, although data presented does support the importance of the identified cation binding site, additional experimental data will be required. Also the manuscript needs significant improvements in presentation.

Major issues

It is argued that site 2 must be a Mg^{2+} ion based on an analysis of ligand-ion distances and coordination, and in the discussion it is mentioned " We modelled the cation at site 2 in our crystal as Mg^{2+} , based on an analysis of bond length, coordination number and behavior of crystallographic refinements with different cations modelled". It would strengthen the manuscript if figures showing omit densities of sites 2 and 3 were displayed along with difference densities for structures with Ca^{2+} or Mg^{2+} modelled. Also we need to know how the cation-ligand distances were restrained during refinement. Was this done automatically by phenix.refine or were additional restraints used?

Related to this issue is also the suggestion reached based on binding data and literature that in

vivo site 2 would be occupied by Ca²⁺. It appears obvious to collect additional diffraction data from crystals soaked in buffers with Ca²⁺ and Mg²⁺ present at physiological relevant concentrations where data is collected at a wavelength in which an anomalous signal will be present if Ca²⁺ and not Mg²⁺ is bound. This is a relatively trivial experiment, but it would substantially strengthen the model of a Ca²⁺ binding site 2.

An ITC (or alternative technique) titration of wt and mutated SRCR 8 with Ca²⁺ and Mg²⁺ may also need to be considered, this would also provide a binding constant for the wt SRCR8 that made it possible to judge the in vivo relevance of Ca²⁺ binding to site 2. Although the data presented in figure 3 supports Ca²⁺ binding as important for ligand binding they do not prove it. In panels A+B protein is eluted with EDTA, but it is not discussed whether EDTA would extract Mg²⁺ from site 1. If that happened, the data do not only report on site 2.

Minor issues

P4-L16 (page 4 line 16). "Ni²⁺-chromatography column (1 ml, Roche)", provide catalogue number P4-L18 "S75 16/60 HR column(GE)". Is this a Superdex 75 from GE Healthcare?

P4-L23 protein was eluted in 250 mM Imidazole. Were there other components in the buffer and what was the pH?

P4-L34 "crystallization buffer was supplemented with 10 mM Mg²⁺ and 10 mM Ca²⁺ 24 hours prior to freezing." Is this a soaking experiment in reservoir buffer?

P5-L1-2. Give product number for the HAP column. The 80 ul SRCR8 added what buffer was the protein in?

P5-L9. What buffer was the heparin column equilibrated in ?

P6-L11. Fig1-> fig 1b

Figure 1B. The Mg²⁺ should have different colors in the two superimposed structures

Table 1. Provide the Mg²⁺ and Ca²⁺ status clearly in the header. Provide the wavelength where the data was collected. Is there any anomalous signal from Ca²⁺ at this wavelength?

Figure 2, line 11. Lowercase "e" -> "E" for legend to panel E

Figure 3. The spy-2 concentration used for coating is given as ug/ml in panel C but in uM in legend.

Be consistent. The remark on "molar excess" in legend to panel C is irrelevant as there is no quantitation of how many molecules of Spy-2 there is bound to surface in each well

The chromatogram in panel B is not of publication quality, as labels are far too small and lines are too thin. Also the area under the three peaks are not the same. Were the amounts of SRCR8 the same in the three experiments.

P11 L5. Reference to FFAS search appears not to be appropriate reference

P13. L24-26 "A search with the SRCR8 model on the Consurf server [44], show that D1019, D1020, D1058 and E1086 all score 7 (out of 9, highly conserved), while N1081 and D1059 score 6 and 4, respectively"

This is a very esoteric statement, only readers familiar with Consurf may grasp the significance.

Please explain better to the broader audience

RE: Life Science Alliance manuscript LSA-2019-00502-T: The structure of SALSA/DMBT1 SRCR domains reveal the conserved ligand-binding mechanism of the ancient SRCR-fold

Dear Editor,

We thank the reviewers for their thoughtful comments about our manuscript. We have attempted to address their concerns in our revision and include a point-by-point description of the changes made below.

While the additional crystallographic experiments suggested by reviewer 2 are theoretically straight-forward, the lead researcher of the study has left the laboratory and the insect cell lines used for producing the protein were shut down. We have therefore re-analysed our existing data and altered our conclusions regarding the likely identity of the metal ions in the binding site of the crystal structure. However, this doesn't change the results of our assays or our opinion on the likely physiological ion. We hope that the improved manuscript is now suitable for publication in Life Science Alliance.

Reviewer #1 (Comments to the Authors (Required)):

The manuscript of Reichhardt et al. describes the structure determination of the SRSC domains 1 and 8 from SALSA/DMBT1 using X-ray crystallography. This protein is involved in recognition of different ligands relevant to mucosal immunology. Assays were carried out to test binding of structurally unrelated ligands, and mutation of residues involved in single and dual binding of calcium atoms (D1019A and D1020A) showed that Ca²⁺ mediates ligand binding.

As pointed out by the authors, these domains have several homologues of known structure and high sequence similarity. Also, the role of structurally equivalent residues involved in calcium binding has been tested previously in these homologues by mutagenesis and binding assays.

I appreciate that the authors carried out SPR binding assays to obtain binding constants of individual domains but observed only weak interactions, and to me the main novelty of the manuscript is in Fig. 3, showing experiments to test

binding against hydroxyapatite, heparin and the Spy-2 domain of Spy0843. Comparison of the binding data of WT and mutant domains demonstrates that calcium is essential for a unified mode of ligand binding, an important point to make considering that other SRSC domains can bind ligands using other regions on the domain surface. In page 14 of the discussion, lines 9-11 it says "... experimental data demonstrated that binding to the ligands tested was only dependent on the presence of Ca²⁺, and not Mg²⁺". To test this, binding assays of the WT SRSC8 domain were carried out in 1mM Ca²⁺ and also in 1mM EGTA (chelator that binds Ca²⁺ with greater affinity than Mg²⁺) and 1mM Mg²⁺. Strictly, I am not convinced that the domain would not bind ligands using Mg²⁺ if that was the only metal available, but their argument that Ca²⁺ levels in the extracellular medium are much higher than those of Mg²⁺ does suggest that Ca²⁺ is more biologically relevant.

The crystallographic data and figures are clearly laid out.

The paper is carefully written and I could not spot any typos.

I recommend publication of the manuscript without any further changes.

We thank the reviewer for their kind comments. We have addressed this issue of Mg²⁺ versus Ca²⁺ binding in more detail in response to reviewer 2.

Reviewer #2 (Comments to the Authors (Required)):

The manuscript by Reichardt et al describes crystal structures of two SRCR domains from SALSA. Based on mutational analysis, in vitro interaction experiments and bioinformatics they propose that a binding site for a divalent metal ion is conserved in many other SRCRs and that they mediate ligand binding. Their results are potentially of significant interest to investigators working with SRCR containing proteins. However, although data presented does support the importance of the identified cation binding site, additional experimental data will be required. Also the manuscript needs significant improvements in presentation.

We thank the reviewer for recognising the significance of the work and for helping us to improve the manuscript.

Major issues

It is argued that site 2 must be a Mg²⁺ ion based on an analysis of ligand-ion distances and coordination, and in the discussion it is mentioned " We modelled the cation at site 2 in our crystal as Mg²⁺, based on an analysis of bond length, coordination number and behavior of crystallographic refinements with different cations modelled". It would strengthen the manuscript if figures showing omit densities of sites 2 and 3 were displayed along with difference densities for structures with Ca²⁺ or Mg²⁺ modelled. Also we need to know how the cation-ligand distances were restrained during refinement. Was this done automatically by phenix.refine or were additional restraints used? Related to this issue is also the suggestion reached based on binding data and literature that in vivo site 2 would be occupied by Ca²⁺. It appears obvious to collect additional diffraction data from crystals soaked in buffers with Ca²⁺ and Mg²⁺ present at physiological relevant concentrations where data is collected at a wavelength in which an anomalous signal will be present if Ca²⁺ and not Mg²⁺ is bound. This is a relatively trivial experiment, but it would substantially strengthen the model of a Ca²⁺ binding site 2.

We thank the reviewer for addressing the specifics of the Calcium versus Magnesium binding of the structure. While we agree that the crystallographic experiments suggested should be trivial, the lead author on the study has left the laboratory and all of the protein stocks were used up in the preparation of this manuscript. As the protein was produced in an insect cell culture system and would require re-establishment of the cell lines, producing fresh protein, and hence crystals, is not a trivial endeavour.

We have, however, re-analysed all of our crystallographic data (including analysing potential anomalous signal – which was not evident). In light of this, we have changed the bound cations in the structure to all be Magnesium – as this was most consistent with the re-refinements and the crystallisation conditions. We have altered the figures and the

discussions in the text to be clearer on this point and introduced Figure S1, displaying the Fo-Fc maps with different cations modelled.

The main argument of this paper is that we identify the presence of this dual cation-binding site and show its relevance for ligand interaction. The specific presence of Magnesium or Calcium at this site will most likely be influenced by specific conditions. Under physiological conditions, where the concentration of Calcium is much higher than that of Magnesium, the physiological ligand will most likely be Calcium. The functional aspects of this study, supported by mutational binding studies, remain unaltered. We have adjusted our argumentation accordingly, and only specify the presence of a cation-binding site, with further functional support for the physiological role of Calcium at this site.

An ITC (or alternative technique) titration of wt and mutated SRCR 8 with Ca²⁺ and Mg²⁺ may also need to be considered, this would also provide a binding constant for the wt SRCR8 that made it possible to judge the in vivo relevance of Ca²⁺ binding to site 2. Although the data presented in figure 3 supports Ca²⁺ binding as important for ligand binding they do not prove it. In panels A+B protein is eluted with EDTA, but it is not discussed whether EDTA would extract Mg²⁺ from site 1. If that happened, the data do not only report on site 2.

ITC experiments were attempted with SRCR domains and various relevant ligands. However, the protein precipitated under all experimental conditions tested. The Spy-2 + WT SRCR8 ELISA was repeated in an MgEGTA buffer. This highlights the necessity of the presence of Calcium.

Minor issues

P4-L16 (page 4 line 16). "Ni²⁺-chromatography column (1 ml, Roche)", provide catalogue number

The catalogue number has been added.

P4-L18 "S75 16/60 HR column(GE)". Is this a Superdex 75 from GE Healthcare?

Yes. The full name has been written out.

P4-L23 protein was eluted in 250 mM Imidazole. Were there other components in the buffer and what was the pH?

The full buffer composition has been written out: 50 mM Tris, pH: 8.5, 200 mM NaCl, 250 mM imidazole.

P4-L34 "crystallization buffer was supplemented with 10 mM Mg²⁺ and 10 mM Ca²⁺ 24 hours prior to freezing." Is this a soaking experiment in reservoir buffer?

Yes.

P5-L1-2. Give product number for the HAP column. The 80 ul SRCR8 added what buffer was the protein in?

Catalogue number has been added for the HAP resin. The protein was diluted in the buffer described in the previous sentence. To clarify this, a sentence was added "in the same buffer".

P5-L9. What buffer was the heparin column equilibrated in ?

The column was equilibrated in the buffer described in the previous sentence. To clarify this a sentence was added: "equilibrated in the same buffer."

P6-L11. Fig1-> fig 1b Figure 1B. The Mg²⁺ should have different colors in the two superimposed structures

1b has been changed to 1B. The Mg²⁺ has been represented in two different colors.

Table 1. Provide the Mg²⁺ and Ca²⁺ status clearly in the header. Provide the wavelength where the data was collected. Is there any anomalous signal from Ca²⁺ at this wavelength?

A sentence was added to the methods section: "Data were collected at a temperature of 80 K on beamlines I04, at a wavelength of 1.0718 Å (for SRCR1 and SRCR8) and I03, at a wavelength of 0.9762 Å (for SRCR8soak) at the Diamond Light Source (Harwell, UK), as specified in Table 1". No anomalous signal was observed.

Figure 2, line 11. Lowercase "e" -> "E" for legend to panel E

Lowercase was altered to uppercase.

Figure 3. The spy-2 concentration used for coating is given as ug/ml in panel C but in uM in legend. Be consistent. The remark on "molar excess" in legend to panel C is irrelevant as there is no quantitation of how many molecules of Spy-2 there is bound to surface in each well

The remark on molar excess has been removed. The concentrations in the figure legends have been altered to ug/ml.

The chromatogram in panel B is not of publication quality, as labels are far too small and lines are too thin. Also the area under the three peaks are not the same. Were the amounts of SRCR8 the same in the three experiments.

The chromatogram has been updated to publication-quality. The amounts of SRCR8 protein loaded was the same, however, there may be slight variations in the loss of material during the loading process.

P11 L5. Reference to FFAS search appears not to be appropriate reference

The reference has been altered.

P13. L24-26 "A search with the SRCR8 model on the Consurf server [44], show that D1019, D1020, D1058 and E1086 all score 7 (out of 9, highly conserved), while N1081 and D1059 score 6 and 4, respectively" This is a very esoteric statement, only readers familiar with Consurf may grasp the significance. Please explain better to the broader audience

To clarify this point, the section on Consurf has been altered to the following: "The Consurf server is a tool to estimate (on a scale from 1-9) the level of evolutionary conservation of residues in a given fold {{801 Ashkenazy,H. 2016;}}. A search with the SRCR8 model show that D1019, D1020, D1058 and E1086 all score 7 (highly conserved), while N1081 and D1059 score 6 and 4, respectively (thus less conserved)."

February 6, 2020

RE: Life Science Alliance Manuscript #LSA-2019-00502-TR

Martin Reichhardt
Oxford, University of
Sir William Dunn School of Pathology
South Parks Road
South Parks Road
Oxford, UK-Oxford OX1 3RE
United Kingdom

Dear Dr. Reichhardt,

Thank you for submitting your revised manuscript entitled "Structures of SALSA domains reveal the conserved ligand-binding mechanism of the ancient SRCR-fold". As you will see, rev#2 appreciates the introduced changes and now supports publication here. We would be happy to publish your paper in Life Science Alliance pending final minor revisions to address reviewer #2's final suggestions. Please also make sure to fill in the electronic license to publish form.

A. FINAL FILES:

B. MANUSCRIPT ORGANIZATION AND FORMATTING:

Sincerely,

Reviewer #2 (Comments to the Authors (Required)):

The revised version has answered in a satisfying manner to the issues raised in the initial review. I

have only minor suggestions for improvements

Page 6, lines 13-14. It would be useful if typical values for Mg^{2+} and Ca^{2+} concentration in tissues where SALSA is present were given here.

Page 9 24, Superdex 75, not S75

P15 line 10 "found to coordinate a metal," -> "found to coordinate a metal ion,"

Figure 4. alignment. It would be useful if the full length numbering for the proteins were given instead of just starting from 1 at each sequence

Fig S1. Label residues on one panel, put legend with Fig S1 and explain the cation coloring

Response to remaining comments from the reviewers.

Reviewer #2 (Comments to the Authors (Required)):

The revised version has answered in a satisfying manner to the issues raised in the initial review. I have only minor suggestions for improvements

Page 6, lines 13-14. It would be useful if typical values for Mg^{2+} and Ca^{2+} concentration in tissues where SALS is present were given here.

- **Values have been added, though on page 7 for better context.**

Page 9 24, Superdex 75, not S75

- **Has been changed.**

P15 line 10 "found to coordinate a metal," -> "found to coordinate a metal ion,"

- **Has been changed.**

Figure 4. alignment. It would be useful if the full length numbering for the proteins were given instead of just starting from 1 at each sequence

- **Has been changed.**

Fig S1. Label residues on one panel, put legend with Fig S1 and explain the cation colouring

- **Panel and legend have been altered.**

February 14, 2020

RE: Life Science Alliance Manuscript #LSA-2019-00502-TRR

Martin Reichhardt
Oxford, University of
Sir William Dunn School of Pathology
South Parks Road
South Parks Road
Oxford, UK-Oxford OX1 3RE
United Kingdom

Dear Dr. Reichhardt,

Thank you for submitting your Research Article entitled "Structures of SALSA domains reveal the conserved ligand-binding mechanism of the ancient SRCR-fold". It is a pleasure to let you know that your manuscript is now accepted for publication in Life Science Alliance. Congratulations on this interesting work.

*****IMPORTANT:** If you will be unreachable at any time, please provide us with the email address of an alternate author. Failure to respond to routine queries may lead to unavoidable delays in publication.*******

DISTRIBUTION OF MATERIALS:

Again, congratulations on a very nice paper. I hope you found the review process to be constructive and are pleased with how the manuscript was handled editorially. We look forward to future exciting

submissions from your lab.

Sincerely,
